# Retention in Care, Mortality, Loss-to-Follow-Up, and Viral Suppression among Antiretroviral Treatment-Naïve and Experienced Persons Participating in a Nationally Representative HIV Pre-Treatment Drug Resistance Survey in Mexico

**DOI:** 10.3390/pathogens10121569

**Published:** 2021-12-01

**Authors:** Yanink Caro-Vega, Fernando Alarid-Escudero, Eva A. Enns, Sandra Sosa-Rubí, Carlos Chivardi, Alicia Piñeirúa-Menendez, Claudia García-Morales, Gustavo Reyes-Terán, Juan G. Sierra-Madero, Santiago Ávila-Ríos

**Affiliations:** 1Departamento de Infectología, Instituto Nacional de Ciencias Médicas y Nutrición Salvador Zubirán, Mexico City 14000, Mexico; yanink.caro@infecto.mx (Y.C.-V.); jsmadero@gmail.com (J.G.S.-M.); 2Centro de Investigación y Docencia Económicas, Aguascalientes 20313, Mexico; fernando.alarid@cide.edu; 3Division of Health Policy and Management, School of Public Health, University of Minnesota, Minneapolis, MN 55455, USA; eens@umn.edu; 4Center for Health Systems Research, Instituto Nacional de Salud Pública, Cuernavaca 62100, Mexico; sandra.sosa.rubi@gmail.com (S.S.-R.); krloschivardi@gmail.com (C.C.); 5Dirección de Atención Integral CENSIDA, Mexico City 14080, Mexico; aliciapina@yahoo.co.uk; 6Centro de Investigación en Enfermedades Infecciosas, Instituto Nacional de Enfermedades Respiratorias, Mexico City 14080, Mexico; claudia.garcia@cieni.org.mx (C.G.-M.); gustavo.reyesteran@gmail.com (G.R.-T.)

**Keywords:** HIV, drug resistance, surveillance, public health, Mexico

## Abstract

We describe associations of pretreatment drug resistance (PDR) with clinical outcomes such as remaining in care, loss to follow-up (LTFU), viral suppression, and death in Mexico, in real-life clinical settings. We analyzed clinical outcomes after a two-year follow up period in participants of a large 2017–2018 nationally representative PDR survey cross-referenced with information of the national ministry of health HIV database. Participants were stratified according to prior ART exposure and presence of efavirenz/nevirapine PDR. Using a Fine-Gray model, we evaluated virological suppression among resistant patients, in a context of competing risk with lost to follow-up and death. A total of 1823 participants were followed-up by a median of 1.88 years (Interquartile Range (IQR): 1.59–2.02): 20 (1%) were classified as experienced + resistant; 165 (9%) naïve + resistant; 211 (11%) experienced + non-resistant; and 1427 (78%) as naïve + non-resistant. Being ART-experienced was associated with a lower probability of remaining in care (adjusted Hazard Ratio(aHR) = 0.68, 0.53–0.86, for the non-resistant group and aHR = 0.37, 0.17–0.84, for the resistant group, compared to the naïve + non-resistant group). Heterosexual cisgender women compared to men who have sex with men [MSM], had a lower viral suppression (aHR = 0.84, 0.70–1.01, *p* = 0.06) ART-experienced persons with NNRTI-PDR showed the worst clinical outcomes. This group was enriched with women and persons with lower education and unemployed, which suggests higher levels of social vulnerability.

## 1. Introduction

HIV pretreatment drug resistance (PDR), particularly to non-nucleoside reverse transcriptase inhibitors (NNRTI) is associated with lower viral suppression (VS) in persons that initiate NNRTI-based antiretroviral treatment (ART) regimens [1,2]. Solid evidence suggests that NNRTI PDR levels have been consistently increasing in low-/middle-income countries (LMICs) worldwide during the last decade [3], posing a significant threat for the achievement of UNAIDS 95–95–95 goals for ending the AIDS epidemic [4]. Mexico is not an exception to this trend, with recent studies showing increasing NNRTI PDR trends in three focal points of the HIV epidemic in the country [5]. A large nationally representative survey performed in Mexico in 2017–2018 showed a PDR level to NNRTI in all ART initiators of 9.9% (95% CI: 8.7–11.2%), ranging from 8.6% (7.4–9.9%) in ART-naïve individuals to 26.2% (19.5–34.3%) in previously antiretroviral-exposed individuals that re-start ART [6,7]. Up until late 2019, Mexican HIV treatment guidelines recommended NNRTI-based first-line ART options and did not recommend the use of routine drug resistance testing before ART initiation, which was instead reserved for cases of documented virological failure [6,7]. However, in 2019, the preferred first line ART options were modified, favoring the use of bictegravir and dolutegravir over efavirenz as the preferred third drug [8]. Since 2014, several LMICs have implemented nationally representative PDR surveys following WHO recommendations [6]. Among 18 countries reporting nationally representative PDR data, 12 showed NNRTI PDR levels over the 10% WHO-recommended threshold to urgently shift to a first-line non-NNRTI-based ART option [6]. Overall, NNRTI PDR levels were observed to be three-times higher among persons with previous exposure to antiretrovirals and two-times higher among women compared to men [6]. On the other hand, recent data on viral suppression at 12 months of ART initiation (defined as a viral load below 1000 copies/mL) in nine countries reporting nationally representative data on acquired drug resistance surveys designed according to WHO recommendations [6,7,8,9], ranged from 72% to 96% [6]. However, considering people not retained in care as virological failures, the prevalence of viral load suppression dropped by 12–22 points [6].

In Mexico, a significantly lower viral suppression among ART-naïve persons with documented PDR has been reported compared to those without PDR [10], but little is known about the association of HIV drug resistance and other outcomes such as retention in care or probability of death. Describing the sociodemographic characteristics, HIV drug resistance prevalence, pre-exposure levels to antiretroviral drugs, retention in care, and virological outcomes of persons initiating ART, could help strengthen HIV programs and support policy making. In this work, using nationally representative data on HIV drug resistance from a previously reported PDR survey [7], together with data from the National HIV Database SALVAR (Mexican System of Distribution, Logistics, and ART Surveillance), we explored longitudinal associations of PDR in persons entering to care and different outcomes such as retention in care, loss to follow-up (LTFU), viral suppression, and death in Mexico.

## 2. Results

### 2.1. Study Population Description

Of 2006 participants with an HIV drug resistance test in the published Mexico PDR survey [7], a total of 1823 (91%), were found in SALVAR and followed for a median of 1.88 years (IQR: 1.59–2.02) and are our study population. Among those, 231 (13%) were classified as ART-experienced and 185 (11%) were resistant to NNRTI. Considering prior exposure to ART and presence of NNRTI PDR, we classified 20 (1%) participants as experienced + resistant; 211 (11%) as experienced + non-resistant; 165 (9%) as naïve + resistant; and 1427 (78%) as naïve + non-resistant. Briefly, 333 (18%) of the study population were females and 1490 (72%) male. The median age was 30 years (IQR: 25–38). Regarding transmission risk, 304 (17%) were heterosexual cisgender women, 1008 (55%) were men who have sex with men (MSM), 326 (18%) were heterosexual cisgender men, 43 (2%) were persons who inject drugs (PWID), and 142 (8%) participants had missing information on transmission risk. Among heterosexual persons, 326 (52%) were cisgender men. The median CD4+ T cell count at the time of HIV drug resistance testing was 229 cells/mm^3^ (IQR: 84–411). Regarding education level, 318 (17%) participants had elementary level or lower, and 1440 (79%) had high school level or higher. Additionally, 888 (51%) participants were employed, 680 (39%) were unemployed, and 160 (9%) were students. A total of 1728 (95%) participants had first ART regimen information, 1136 (66%) of them based on EFV. Clinical and sociodemographic characteristics of patients by group are shown in Table 1.

The percentage of ART-experienced individuals was higher among women and heterosexual men (70/333; 21% and 39/326, 12%, respectively) compared to MSM (88/1008, 9%; *p* < 0.01). Considering both persons with prior ART exposure and ART-naïve persons, the prevalence of resistance to NNRTI among women (41/333; 12%) and among heterosexual men (40/326, 12%) was higher than among MSM (87/1008, 8.6%; *p* = 0.05).

### 2.2. Characteristics of Participants without Information in the National HIV Database 

A total of 184 (9.2%) persons were not included in the study because they were not found in the SALVAR dataset. Of them, 182 (99%) were naïve to ART, and 18 (10%) had NNRTI resistance, all of them belonging to the naïve group. When compared to those with available information in the SALVAR dataset, 17 (9.4%) were cisgender women (*p* < 0.01), the median age was 28 years (IQR: 24–37; *p* = 0.21), the median CD4 cell count was 287 cells/mm3 (IQR: 124–419; *p* = 0.11), 87% had high school level or higher education (*p* = 0.02), and 61% were employed (*p* < 0.01).

### 2.3. Final Outcomes

Considering 1823 persons with an HIV drug resistance test and information available in SALVAR, the present study represented 3034.3 person-years of follow-up. At the end of follow-up, 1276 (70%) of participants were reported as “in care” in SALVAR, 102 (6%) were reported as dead, and 435 (24%) were reported as LTFU. Among participants classified as LTFU, no specific reason was registered for 206 (46%), while 147 (33%) reported a change to a different health system (mainly due to employment status change and acquisition of social security), 82 (18%) left care for other reasons, and 10 (2%) discontinued ART. The distribution of final outcomes by group was non-significantly different (*p* = 0.06); however, we observed a higher proportion of participants retained in care among ART naïve persons (71%), compared to ART-experienced persons (63%, *p* = 0.002). Importantly, a trend toward higher mortality was observed in the experienced + resistant group (15%) compared to the experienced + non-resistant (8%), the naïve + non-resistant (5%), and the naïve + resistant (4%; *p* = 0.08) groups (Figure 1).

#### 2.3.1. Viral Suppression

Viral load data in the last six months of follow-up was available for 1637 (89%) participants, among whom 1126 (68%) had achieved viral suppression. When comparing across groups, 51% (92/179) among experienced + non-resistant; 36% (5/14) among experienced + resistant; 72% (929/1294) among naïve + non-resistant; and 67% (100/150) among naïve + resistant, achieved viral suppression (*p* < 0.001). Of the 1637 individuals with viral load follow up data available, 1259 (77%) were recorded as still in care at the end of follow-up, with 1021 (81%) of them achieving viral suppression; 330 (26%) classified as LTFU; and 48 (3%) as dead. After multivariable adjustment, experienced + non-resistant participants (aOR = 0.46, 95% CI: 0.32–0.66) and experienced + resistant (aOR = 0.28, 95% CI: 0.09–0.87) had lower odds of viral suppression compared to the naïve + non-resistant group. (Table 2, Model 1). Note that, in this analysis, 30% of the experienced + resistant group was not included due to lack of follow-up viral load data (Appendix A). In the analysis using multiple imputation we observed significantly lower odds of viral suppression in experienced + non-resistant (aOR = 0.49, CI95%: 0.34–0.70), but not in experienced + resistant (aOR = 0.39, 95% CI: 0.13–1.15), and naïve + resistant (aOR = 0.79, 95% CI: 0.53–1.17) compared to naïve + non-resistant participants (Table 2, Model 2). Moreover, older participants had higher odds of viral suppression (aOR = 1.45, 95% CI: 1.04–2.04, for 50 years old vs. 30 years old) in the first model, but not in second with the imputed data set (aOR = 1.29, 95% CI: 0.93–1.79) (Table 2, Model 2). 

#### 2.3.2. Change in Antiretroviral Treatment Regimen

Of the 1136 (66%) participants who started ART with EFV-based regimens, 907 (80%) of them belonged to the naïve + non-resistant group, 9 (<1%) to the experienced + resistant, 100 (9%) to the naïve + resistant, and 120 (10%) to the experienced + non-resistant group. ART-experienced participants were more likely to switch to NNRTI-sparing regimens, with 40% of the non-resistant and 20% of the resistant. By contrast, within the ART naïve participants, 6% of the non-resistant, and 5% of the resistant changed ART regimen. When including information regarding change in ART regimen in the logistic model, the odds of viral suppression was significantly higher in persons who changed versus those who did not change regimen (aOR = 1.78, 95% CI: 1.15–2.75; *p* < 0.01). The odds of viral suppression for experienced + non-resistant (aOR = 0.37, 95% CI: 0.24–0.53) and experienced + resistant (aOR = 0.26, 95% CI: 0.08–0.83) compared to naïve + non-resistant persons including data on change in ART regimen, remained similar to those of the previous model (Table 2, Model 3).

#### 2.3.3. Viral Suppression Outcome with Lost to Follow-Up and Death as Competing Events

Among the 1637 participants with viral load data available, we found that 1021 persons (62%) ended the study follow-up in care and virally suppressed, 238 (14%) were in care but without viral suppression, 330 (20%) were LTFU, and 48 (3%) were reported as dead. By group, the highest proportion of participants classified as in care and suppressed was observed among ART-naïve participants (84% among non-resistant and 77% among resistant), compared with ART-experienced participants (64% among non-resistant and 50% among resistant) (Appendix A). Using a Fine-Gray model, we found that being ART-experienced was associated with a lower probability of remaining in care with viral suppression over time (aHR = 0.68, 95% CI: 0.53–0.86, for the non-resistant group and aHR = 0.37, 0.17–0.84, for the resistant group, compared to the naïve + non-resistant group). Older age and higher education level did not show a significant association with viral suppression. Heterosexual cisgender women compared to MSM, had a lower hazard of viral suppression (aHR = 0.84, 95% CI: 0.70–1.01, *p* = 0.06) (Table 3). The estimated probability of remaining in care and virally suppressed, being LTFU and death, over time, for each group is shown in the Appendix A.

## 3. Discussion

The present study provides evidence that ART re-starters, as well as persons with pre-treatment NNRTI resistance, in general have worse clinical outcomes than persons without previous exposure to ART and persons without NNRTI resistance in a cohort of Mexican individuals followed for two years. In the adjusted model with ART-regimen change variable, prior exposure to ART was strongly associated with poor clinical outcomes, and with ART-experienced participants, having nearly 40% lower probability of remaining in care with viral suppression. Previous exposure to ART was more common in women with low education arriving late to clinical care. From a total of 1823 Mexican persons living with HIV with baseline drug resistance testing and followed by a median of almost 2 years of follow-up time, we classified 1% as experienced + resistant, 9% as naïve + resistant, 11% as experienced + non-resistant, and 78% as naïve + non-resistant. At the end of follow-up, the experienced + resistant group had the lowest proportion of participants remaining in care compared to other groups. Participants in the experienced + resistant group also showed higher mortality and LTFU, as well as lower levels of viral suppression, even after switching to NNRTI-sparing ART regimens. 

Participants in the experienced + resistant group were more frequently women, with lower education level and more frequently unemployed, compared to other groups. Higher social and economic vulnerability among women living with HIV in Mexico has been observed in previous studies [10,11,12,13,14,15], including in circumstances associated with HIV diagnosis during pregnancy [15]. Additionally, a higher risk for ART discontinuation has been observed among Mexican women living with HIV compared to men [16], with a recent study reporting as high as 20% ART interruption rate [16], with reasons including access to care, depression, and ART adverse effects. Other common reasons observed for ART interruption, and higher viral failure rate in women include a general lower education level, economic dependence on other members of the household, responsibility to provide care and sustenance for children, long distance to clinics and difficulty to pay for transport [12,15]. In general, we found that MSM had better outcomes than heterosexual participants. This could reflect both significant structural differences in the MSM population compared to the heterosexual population living with HIV in Mexico [12], as well as a traditional focus on MSM as a priority group for HIV care and focus of most national HIV prevention efforts.

Changes in ART regimen were more commonly observed among ART-experienced than naïve patients, regardless of their resistance status. However, from this study, we cannot deduce whether clinicians were more proactive in changing ART regimens for ART-experienced and/or NNRTI resistant patients. We also do not know which information clinicians had available to them at the time of ART-regimen decision-making, as resistance test results. Thus, it is not possible to assess whether the results influenced selection of first-line ART regimens. 

In Mexico, as of 2019, bictegravir, an integrase inhibitor drug with high genetic barrier to resistance [8], is recommended as the basis for first line ART-regimens. However, the use of bictegravir is contraindicated in patients using rifampicin or rifabutin; and in pregnant women. In 2020, 30% of patients were still on EFV-based regimens in Mexico [16]. Thus, the implementation of measures to improve adherence and prevent failures due to resistance are important and still needed. HIV drug resistance testing is recommended to guide the choice of second-line ART line, and adherence issues and potential drug interactions need to be addressed. 

Our study provides an evaluation of clinical outcomes in real-life setting evaluating clinical outcomes among ART-naive and experienced persons with and without resistance to NNRTI, using surveillance data cross-referenced with the official ministry of health database analyzed with robust statistical techniques. We evaluated virological success among resistant patients, in a context of competing risk with lost to follow-up and death; including the effect of ART change and multiple imputation analyses to address possible bias due to missing data. However, our study also has limitations. First, the original surveillance study was not designed to follow-up participants or to evaluate their clinical outcomes in longer periods, which may reduce availability of information. Using the national database SALVAR allowed us to improve completeness of the data, but we acknowledge that some quality issues could exist with a possible impact on our results. The SALVAR database was designed to record viral load and CD4 T cell count studies, as well as antiretroviral drug dispensation practices for administrative purposes, and record of visits to the clinic or vital status are not among its main objectives. We used the information available as a proxy to inform retention in care, lost to follow-up and viral suppression. Second, not all participants of the original surveillance study were included due to lack of follow-up information, representing a possible selection bias. This situation is frequent in the Mexican setting due to the fragmentation of the health system, obligating persons with formal employment to seek clinical care and ART in social security clinics and persons without formal employment in ministry of health clinics. The lack of a unified national database and migration of persons between health systems, due to changes in employment status has been previously reported as an important reason for ART defaulting and LTFU [17]. The percentage of participants in our study in this situation was 9%, and we found some sociodemographic differences between the persons excluded due to missing data and the persons with data available in the SALVAR database, and thus included in the current study, raising representativeness issues. Nevertheless, these participants with missing data were mainly men, with a slightly higher median CD4+ count, with higher education, higher employment rates, and lower prevalence of re-starters. Interestingly, a Mexican study including outpatients of a large Ministry of Health clinic in Mexico City observed that persons arriving to care with CD4+ T cell counts < 100 cells/mm^3^ were more frequently classified as intermittent ART users and 43% came from social security clinics [18], possibly suggesting a return to care in Ministry of Health clinics, in a worse condition after employment loss, ART defaulting, and LTFU. A program linking the different health care systems in Mexico and a unified HIV database are urgently needed to improve follow-up and care of patients repeatedly changing employment status. Finally, although our study leverages access to SALVAR, we recognize that the information available in this national database is limited. The variables collected for sociodemographic description of the study cohort only describe education level and employment, which may also be variable over time. Inclusion of more adequate variables to describe economic class or poverty level could help to better define socioeconomic status and risk associations with the evaluated outcomes.

Our study provides evidence on associations between pretreatment drug resistance and prior exposure to antiretrovirals in persons starting ART and deleterious clinical outcomes in the Mexican context. However, these observations may also be generally true elsewhere, mainly in other LMICs, and help in public health decision making. 

## 4. Materials and Methods

### 4.1. Data Source

We analyzed PDR and sociodemographic data from 1823 persons that participated in a large nationally representative HIV PDR survey carried out in Mexico from 09/2017 to 03/2018 who entered to care in Ministry of Health clinics [7], and had follow-up data in Mexico’s national Ministry of Health ART database (SALVAR). The SALVAR database comprises information regarding mortality, retention in HIV care, ART regimen history, and CD4+ T cell count, and viral load follow-up for persons without social security in Mexico. The administrative closure date for the dataset used in this study was 12/2019. The last status reported in the database was used to classify the final outcome of the participants. 

### 4.2. Sample Selection

Persons recorded as alive and in follow-up in the SALVAR database were classified as remaining in care. Persons with a non-active status due to ART abandonment, migration to other healthcare systems [18], unknown status, as well as lack of viral load follow-up for more than 6 months at the dataset closure date, were classified as LTFU. Persons who died were not included in the LTFU group. Participants were stratified into four groups according to prior ART exposure and presence of NNRTI-PDR: experienced + resistant, experienced + non-resistant, naïve + resistant, and naïve + non-resistant. Retention in care, LTFU and death were compared between groups as final outcomes.

### 4.3. Statistical Analysis

We estimated the percentage of persons with viral suppression (last viral load < 200 copies/mL) at the end of follow-up by group, among those persons with viral load information available (within 6 months prior to the last database entry for each participant). A logistic regression model was developed to assess the relationship between ART exposure, NNRTI resistance, and viral suppression, adjusting for sociodemographic characteristics in the main analysis. A model including change in ART status after an HIV drug resistance test was performed to see the potential impact of this variable on the probability of reaching viral suppression. A Fine-Gray model was used to compare retention in care and viral suppression at the end of follow-up, with LTFU and death as final outcomes and competing events. In this model, we censored persons who ended in care but did not achieve viral suppression or did not have viral load information available. We included ART exposure drug resistance status groups, age, sex, mode of transmission, and educational level as co-variables. Due to the small size of groups, we combined sex and mode of HIV transmission to include them in the models. The variable “sex and mode of transmission” was categorized as: MSM, heterosexual cisgender men and heterosexual cisgender women. People who inject drugs were excluded from the models. Additional analyses using multiple imputations with 10 replications were conducted for viral suppression since we observed that 10% of the records had missing information for the last viral load. In particular, 30% (6/20) of the experienced + resistant group had missing viral load data. Additionally, to evaluate potential biases and the possible generalization of our results, we described and compared the characteristics of HIV drug resistance testing between persons with and without information in SALVAR. All the analyses were performed using R Version 1.2.5019. 

## Figures and Tables

**Figure 1 pathogens-10-01569-f001:**
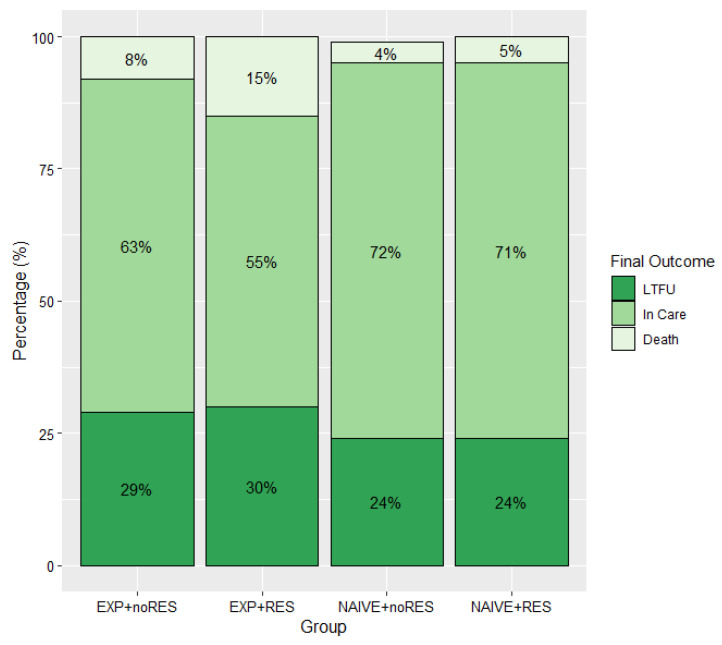
Final outcome by presence of efavirenz/nevirapine pre-treatment drug resistance and prior exposure to antiretroviral drugs in a cohort of Mexican persons living with HIV, 2017–2019. Note: Groups according to prior ART exposure and presence of NNRTI-PDR: EXP + noRES: experienced + non-resistant, EXP + RES: experienced + resistant, NAÏVE + noRES: naïve + non-resistant and NAÏVE + RES: naïve + resistant LTFU, lost to follow-up: defined as persons with a non-active status due to ART abandonment, migration to other healthcare systems, unknown status, as well as lack of viral load follow-up for more than 6 months at the dataset closure date.

**Table 1 pathogens-10-01569-t001:** Clinical and sociodemographic characteristics according to prior exposure to antiretroviral treatment and presence of efavirenz/nevirapine pretreatment drug resistance in Mexican individuals living with HIV, 2017–2019, N = 1823.

	Experienced -Resistant N = 20	Experienced -Non-Resistant N = 211	Naïve -Resistant N = 165	Naïve -Non-Resistant N = 1427	*p*-Value ^1^
Female; n (%)	9 (45%)	69 (33%)	32 (19%)	223 (16%)	<0.01
Median age (years); (IQR)	34 (28–39)	34 (27–42)	30 (25–41)	29 (25–38)	<0.01
Transmission risk *; n (%) Heterosexual cisgender women MSM Heterosexual cisgender men PWID	8 (40%) 5 (20%) 5 (20%) 1 (0.5%)	62 (29%) 83 (39%) 34 (16%) 13 (6%)	30 (18%) 82 (49%) 35 (21%) 3 (1.8%)	204 (14%) 838 (59%) 252 (18%) 26 (1.8%)	< 0.01
Mean CD4+ T cell count; cells/mm^3^ (IQR)	223 (58–410)	143 (53–343)	244 (94–459)	237 (91–413)	0.31
Education; n (%) Elementary or lower High school or higher Unknown	8 (40%) 12 (60%) 0 (0%)	62 (29%) 141 (67%) 8 (4%)	31 (19%) 128 (77%) 6 (4%)	217 (15%) 1159 (81%) 51 (3.5%)	<0.01
Occupation; n (%) Employed Unemployed Student	4 (20%) 16 (80%) 0 (0%)	81 (40%) 107 (53%) 15 (7%)	88 (51%) 63 (37%) 21 (12%)	824 (54%) 532 (35%) 154 (10%)	<0.01
Median time of follow-up (years); (IQR)	1.93 (0.85–2.08)	1.91 (1.51–2.04)	1.86 (1.66–2.05)	1.87 (1.60–2.01)	0.74
First ART regimen group; n (%) Based on EFV Integrase Inhibitors Protease Inhibitors Other	9 (45%) 3 (15%) 7 (35%) 1 (5%)	120 (58%) 15 (7%) 68 (33%) 4 (2%)	100 (64%) 50 (33%) 4 (2%) 1 (1%)	907 (67%) 366 (27%) 70 (5%) 3 (0.2%)	<0.01

^1^ The *p*-value compares the distribution of variables in each group, from Kruskal–Wallis, chi-squares, or fisher test according with the type of variable. * PWID: people who inject drugs includes 43 participants, 39 of them men and 4 women; 26 in the naïve + non-resistant group; 13 in the experienced + non-resistant; 1 in the experienced + resistant; and 3 in the naïve resistant. The missing information for risk of transmission was 1 for experienced + resistant, 19 in the experienced +non-resistant, 107 in the naive-resistant and 15 in the naïve + non-resistant (n = 142).

**Table 2 pathogens-10-01569-t002:** Factors associated with viral suppression in a cohort of Mexican persons living with HIV, 2017–2019.

	Model 1		Model 2		Model 3	
Characteristics	OR; IC95%	*p*-Value	OR; IC95%	*p*-Value	OR; IC95%	*p*-Value
**Group**						
Naïve + non-resistant	1		1		1	
Naïve + resistant	0.74; 0.50–1.09	0.14	0.79; 0.53–1.17	0.25	0.71; 0.47–1.07	0.10
Experienced + non-resistant	0.46; 0.32–0.66	<0.001	0.49; 0.34–0.70	<0.001	0.37; 0.24–0.53	<0.001
Experienced + resistant	0.28; 0.09–0.87	0.02	0.39; 0.13–1.15	0.09	0.26; 0.08–0.83	0.02
**CD4+ T cell count at the time of HIV drug resistance test (cells/mm^3^)**		0.16		0.27		0.04
100	1		1		1	
200 300 400	1.04; 0.89–1.21 1.03; 0.82–1.31 0.98; 0.77–1.25		1.04; 0.90–1.21 1.05; 0.84–1.32 1.00; 0.78–1.28		0.99; 0.85–1.16 0.95; 0.74–1.22 0.88; 0.67–1.14	
**Age at the time of HIV drug resistance test (years) ^1^**		0.17		0.77		
30	1		1		1	0.03
40	1.25; 0.89–1.48		1.09; 0.97–1.24		1.10; 0.84–1.44	
50	1.45; 1.04–2.04		1.29; 0.93–1.79		1.52; 1.05–2.19	
**Transmission Risk**						
MSM	1		1		1	
Heterosexual cisgender men	0.77: 0.56–1.06	0.11	0.80; 0.58–1.07	0.13	0.75; 0.54–1.05	0.09
Heterosexual cisgender women	0.74; 0.53–1.03	0.82	0.73; 0.53–1.00	0.69	0.76; 0.53–1.08	0.97
**Education level**		0.56		0.73		
Elementary or lower	0.90; 0.64–1.27		0.94; 0.68–0.30		0.89; 0.62–1.28	0.55
High school or higher	1		1		1	
**Employment status**		0.73		0.99		
Unemployed	0.96; 0.75–1.22		1.00; 0.79–1.27		0.76; 0.59–0.99	0.004
Employed	1		1		1	
**Change in ART regimen**	NA	NA	NA	NA	1.78; 1.15–2.75	0.009

Model 1 was fitted by sex, antiretroviral treatment + exposure drug resistance group, CD4+ T cell count, age, transmission risk, education level and employment status, n = 1454; Model 2 includes the same variables of Model 1 using imputation of missing data to improve dataset completeness, n = 1780; and Model 3 includes the variable change in the ART regimen, n = 1445. ^1^ Age was modelled using splines with 3 nodes, the reference age selected for the results was 30 years old. MSM: Men who have sex with men, NA: not available.

**Table 3 pathogens-10-01569-t003:** Variables associated to the probability of viral suppression at the end of follow-up in a competing risk context. (Fine-Gray model).

	HR ^1^	95% CI	*p*-Value
**Group**			
Naïve + non-resistant	1		
Naïve + resistant	0.97	(0.78–1.22)	0.81
Experienced + non-resistant	0.68	(0.53–0.86)	<0.001
Experienced + resistant	0.37	(0.17–0.84)	0.01
**CD4+ T cell count at the time of HIV drug resistance test (per 100 cells/mm^3^)**	0.99	(0.96–1.03)	0.78
**Age at the time of HIV drug resistance test**	1.03	(0.97–1.10)	0.40
**Transmission risk**			
MSM	1		
Heterosexual cisgender women	0.84	(0.70–1.01)	0.06
Heterosexual cisgender men	0.91	(0.76–1.09)	0.29
**Education level**			
Elementary or lower	1		
High school or higher	1.14	(0.99–1.31)	0.07

^1^ HR: hazard ratio, 95% CI: 95% confidence interval, MSM: men who have sex with men.

## Data Availability

The data presented in this study are available on request from the corresponding author. Databases with patient sequences and follow up data are not public due to national regulations.

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
