# Peer review of "Retention in Care, Mortality, Loss-to-Follow-Up, and Viral Suppression among Antiretroviral Treatment-Naïve and Experienced Persons Participating in a Nationally Representative HIV Pre-Treatment Drug Resistance Survey in Mexico"

_pathogens, 2021, doi:10.3390/pathogens10121569_

Round 1
Reviewer 1 Report
It's a good piece of work that was a pleasure to read.
There are a number of remarks and questions that arose in the course of reading.
Remarks:
• The acronym SALVAR occurs for the first time on page 2, but only deciphers on page 8,
• The sentence on page 2 (lines 86-88) is difficult to understand. In one sentence, the authors talk about both the sex and the transmission routes. Moreover, the sum of all patients with known transmission routes is 1681, not 1823. Only further in the text it becomes clear that for the rest the transmission route is unknown,
• Page 4, Figure 1. The columns in the diagram are presented disproportionately to their values
Questions:
• It is not mentioned how long the patients took ART before taking the viral load test. Is this information known? Because it can significantly affect the number of people who reach undetectable viral load,
• Did the investigators have information on the date of diagnosis and whether the relationship between duration of infection and LTFU was analyzed?
Author Response
Remarks:
• The acronym SALVAR occurs for the first time on page 2, but only deciphers on page 8,
Response: Thanks, we added the description of SALVAR on page 2. SALVAR: Mexican System of Distribution, Logistics and ART Surveillance
The sentence on page 2 (lines 86-88) is difficult to understand. In one sentence, the authors talk about both the sex and the transmission routes. Moreover, the sum of all patients with known transmission routes is 1681, not 1823. Only further in the text it becomes clear that for the rest the transmission route is unknown,
Response: Many thanks for pointing this out. We changed the complete sentences and added more information about the distribution of transmission risk, including the percentages relative to the total data and the missing values.
Page 4, Figure 1. The columns in the diagram are presented disproportionately to their values
Response: We appreciate the thorough review. We found other typos in other labels and remade the figure.
Questions:
• It is not mentioned how long the patients took ART before taking the viral load test. Is this information known? Because it can significantly affect the number of people who reach undetectable viral load,
Response: We agree with the reviewer and understand their point. In preliminary analysis, which is not included in the manuscript, we evaluated only participants with ART information available and those with a viral load after 6 months of the ART initiation date. We found 1637 participants with viral load measurement in the last 6 months of follow-up, independently of having ART information or the time between ART date initiation and viral load measurement, and classified 511 as non-suppressed and 1126 as suppressed. If we include the additional conditional time between ART and viral load dates, we will lose 963 participants from the denominator, 674 with an undetectable viral load. In consequence, with this approach, we would have left many people out of the denominators. We prefer to report our current measure of viral suppression, although we know that other definitions are more commonly used.
- Did the investigators have information on the date of diagnosis and whether the relationship between duration of infection and LTFU was analyzed?
Response: Unfortunately, we don´t have information about HIV diagnosis to evaluate its potential association with LTFU.
Reviewer 2 Report
This work is a retrospective real-life cohort study derived from a large survey of HIV pre-treatment drug resistance evaluating the impact of such resistance on both previously naive patients and those experienced in relevant outcomes such as retention in care, loss of follow-up, virological suppression, and death. This analysis makes the final findings significant, especially in countries and regions where first-line NNRTI and protease inhibitors-based therapies are still prevalent.
Some observations to make are:
- SALVAR, the nationwide system from which the data are obtained, essentially brings together the population living with HIV from the public health sector of Mexico, with different risk profiles, adequately evaluated by the Fine-Gray model used. However, the socioeconomic variables reported in the manuscript are only related to education level and employment. Although they are fundamental, they are not absolute nor exclusive; for one example, employment status is variable over time. Therefore, having included the variable of economic class or poverty level would have better helped define socioeconomic risk in the evaluated outcomes or establish it as a limitation if it was not available.
- Given that the population being evaluated in this study comes from a 2018 national PDR survey, it is pertinent to clarify whether these results influenced selecting the first antiretroviral treatment initiated in patients in the study cohort. The inclusion of 165 naive-resistant patients in the follow-up cohort raises this uncertainty.
- Did the authors consider correlating the outcomes assessed with each of the antiretroviral therapeutic classes? Considering that the study's objective is to explore potential associations of PDR in people entering medical care with the evaluated outcomes, it seems relevant to establish the association of initial treatment, the presence of PDR within these groups, and the outcomes.
- Authors should avoid using the terms viral suppression and undetectability as interchangeable, as noted in line 200, given that the threshold used for the study was 200 copies/mL, certainly not equal to undetectable.
- I suggest the authors review the construction of figure 1; the height of the different fractions in the columns does not correspond to the percentages represented in it according to the scale of the Y-axis. Alternatively, explain it.
Author Response
- SALVAR, the nationwide system from which the data are obtained, essentially brings together the population living with HIV from the public health sector of Mexico, with different risk profiles, adequately evaluated by the Fine-Gray model used. However, the socioeconomic variables reported in the manuscript are only related to education level and employment. Although they are fundamental, they are not absolute nor exclusive; for one example, employment status is variable over time. Therefore, having included the variable of economic class or poverty level would have better helped define socioeconomic risk in the evaluated outcomes or establish it as a limitation if it was not available.
Response: We agree with the reviewer that our variables are not enough to completely describe the risk of poor outcomes. However, we think our study is a first attempt to underscore the situation in Mexico and maybe also to guide future studies on this topic. We added a sentence in the discussion section to recognize this point as a limitation on page 9, lines 317-322 of the updated manuscript and reads:
“Finally, although our study leverages access to SALVAR, we recognize that the information available in this national database is limited. The variables collected for sociodemographic description of the study cohort only describe education level and employment, which may also be variable over time. Inclusion of more adequate variables to describe economic class or poverty level could help to better define socioeconomic status and risk associations with the evaluated outcomes.”
- Given that the population being evaluated in this study comes from a 2018 national PDR survey, it is pertinent to clarify whether these results influenced selecting the first antiretroviral treatment initiated in patients in the study cohort. The inclusion of 165 naive-resistant patients in the follow-up cohort raises this uncertainty.
Response: Thanks for the comment. We address this point in the last paragraph, page 8, of the discussion section. We stated that we don´t know if clinicians could be biased in selecting antiretroviral treatment because we don´t know when they received the resistance test results. We reworded the sentence to clarify that all the participants, naive and non-naive, could have received specific ARV regimens because their participation in the resistance study even if clinicians were not aware of the resistant status of each patient.
- Did the authors consider correlating the outcomes assessed with each of the antiretroviral therapeutic classes? Considering that the study's objective is to explore potential associations of PDR in people entering medical care with the evaluated outcomes, it seems relevant to establish the association of initial treatment, the presence of PDR within these groups, and the outcomes.
Response: We agree with the reviewer the potential association of ARV regimen with outcomes is relevant. For this reason, we described the first ARV regimen by resistance and ART status groups (Table 1), and the association of ARV group change with viral suppression (Table 2, model 3). We also included section 2.3.2 Change of antiretroviral treatment regimen in the results. We wanted to conduct more detailed analyses for ARV regimen group, but we think the association could be confounded because of the few participants in the possible combination of ARV regimen and PDR+ARV status groups. For example, in our big group of Naive+non-resistant patients, we have 70 patients initiating with protease Inhibitor regimens, and 11 participants received a change of treatment. Among those, 5 got switched to Bictegravir, 4 to EFV, and 2 to another Integrase regimens. Among those 9 were recorded as reaching viral suppression, and 2 were LTFU at end of the follow-up. We think this descriptive information could be valuable but could not allow us to conclude real and significant associations.
- Authors should avoid using the terms viral suppression and undetectability as interchangeable, as noted in line 200, given that the threshold used for the study was 200 copies/mL, certainly not equal to undetectable.
Response: Thank you, we reviewed the manuscript thoroughly, found the issue, and corrected it in the discussion section and the supplementary Figure S1. We now use the term “viral suppression”.
- I suggest the authors review the construction of figure 1; the height of the different fractions in the columns does not correspond to the percentages represented in it according to the scale of the Y-axis. Alternatively, explain it.
Response: Many thanks for the careful review. We found some labels were wrong in Figure 1 and regenerated them.
Reviewer 3 Report
The authors report a statistical association between pre-treatment drug resistance and clinical outcomes in a large Mexican national HIV cohort. They identify vulnerable groups (women, unemployed, low education) that statistically appear to have worse clinical outcomes compared to other groups. The manuscript is a simple report describing patterns, numbers, and associations in the data. The impact comes from directly dealing with public health records (not easily accessible) and can potentially impact local policy. The conclusions may generally be true for other nations as well, especially LMIC. Perhaps, the authors may add that these results may also be true elsewhere in Discussion (although they do not have access to other countries data). I have no quarrels with the paper.
Author Response
The authors report a statistical association between pre-treatment drug resistance and clinical outcomes in a large Mexican national HIV cohort. They identify vulnerable groups (women, unemployed, low education) that statistically appear to have worse clinical outcomes compared to other groups. The manuscript is a simple report describing patterns, numbers, and associations in the data. The impact comes from directly dealing with public health records (not easily accessible) and can potentially impact local policy. The conclusions may generally be true for other nations as well, especially LMIC. Perhaps, the authors may add that these results may also be true elsewhere in Discussion (although they do not have access to other countries data). I have no quarrels with the paper.
Response: Many thanks for your comments and time. We also think that our results could be helpful to understand the resistance characteristics and outcomes for people in other countries. We have added a comment in the Discussion stating this, as the Reviewer suggests. Please see page 10, lines 344-347.
"Our study provides evidence on associations between pretreatment drug resistance and prior exposure to antiretrovirals in persons starting ART and deleterious clinical outcomes in the Mexican context. However, these observations may also be generally true elsewhere, mainly other LMICs, and help in public health decision making"